# Enterocins: Classification, Synthesis, Antibacterial Mechanisms and Food Applications

**DOI:** 10.3390/molecules27072258

**Published:** 2022-03-30

**Authors:** Yajing Wu, Xinxin Pang, Yansha Wu, Xiayu Liu, Xinglin Zhang

**Affiliations:** 1Department of Food Science and Nutrition, Zhejiang University, Hangzhou 310058, China; wuyajing@zju.edu.cn (Y.W.); 21713048@zju.edu.cn (X.P.); yanshawu@zju.edu.cn (Y.W.); xiayuliu@zju.edu.cn (X.L.); 2College of Agriculture and Forestry, Linyi University, Linyi 276005, China

**Keywords:** enterocins, classification, antimicrobial mechanisms, applications

## Abstract

Enterococci, a type of lactic acid bacteria, are widely distributed in various environments and are part of the normal flora in the intestinal tract of humans and animals. Although enterococci have gradually evolved pathogenic strains causing nosocomial infections in recent years, the non-pathogenic strains have still been widely used as probiotics and feed additives. *Enterococcus* can produce enterocin, which are bacteriocins considered as ribosomal peptides that kill or inhibit the growth of other microorganisms. This paper reviews the classification, synthesis, antibacterial mechanisms and applications of enterocins, and discusses the prospects for future research.

## 1. Introduction

Enterococci are catalase-negative, gram-positive cocci that have an average GC content of less than 40% and produce lactic acid as the main end product of glucose fermentation. *Enterococcus* is a type of facultative anaerobic bacteria that can tolerate temperatures of 10–45 °C and grow best at 35 °C [1]. Enterococci are widely distributed in the natural environment and are part of the normal flora in the intestinal tract of humans and animals. However, in recent years, the bacteria has gradually evolved into a pathogenic and multi-drug resistant branch, and has become an important pathogen that threatens the health of human beings, especially hospitalized patients [2,3]. The drug-resistant *E. faecium* is a member of the so-called “ESKAPE” classification (*E. faecium, Staphylococcus aureus, Klebsiella pneumoniae, Pseudomonas aeruginosa* and *Enterobacter* spp.), which is listed by the Infectious Diseases Society of America (IDSA) as an essential factor in global medical treatment difficulty and death [4].

Previous studies have shown that pathogenic drug-resistant bacteria were significantly more prevalent in hospitalized patients, doctors and nurses than healthy community populations. The major cause of infection with drug-resistant enterococci is its colonization and unbalanced proliferation in the intestine [5]. In recent years, multidrug-resistant enterococci isolated from animal samples have become increasingly frequent [6,7,8], and animal-derived drug-resistant enterococci have higher pathogenic potential and transmission threat than healthy population isolates [9,10]. Farmed animals and pets, as hosts of *Enterococcus*, play an important role in the accumulation of evolutionary drug resistance and pathogenic factors and the transmission between humans and animals. At the same time, intensive and large-scale farming methods have emerged because traditional animal production methods can no longer meet the growing human demand for animal food such as meat, eggs and milk. Under this situation, the immunity of animals is weakened due to the excessive breeding density, and the abuse of antibiotics leads to a great increase in the probability of animals carrying pathogenic and drug-resistant *Enterococcus*, making farmed animals become an important carrier and source of the pathogenic bacteria. Therefore, the use of non-antibiotic methods to specifically reduce or eliminate pathogenic drug-resistant *Enterococcus* carried in the host, while avoiding the development of new drug resistance, will be an ideal method to prevent and treat the disease.

Unlike other pathogenic bacteria, certain non-pathogenic *Enterococcus* isolates are also used as food probiotics or clinical therapeutic probiotics for humans and pets [11,12], and as feed additives for disease control and growth promotion in the breeding industry [13]. This is because many strains of *Enterococcus* can produce bacteriocins, which are known as enterocins.

Bacteriocins are peptide antibiotics secreted by *Enterococcus* during the metabolic process. They are extracellular secreted proteins with certain bacteriostatic activities, and are usually encoded by plasmid genes or chromosomal genes and synthesized by ribosomes [14]. Most bacteriocins only have damaging effects on closely related bacteria, but there are still parts of them with inhibitory effects that are not limited to homologous bacteria. The bacteriocin-producing strain is immune to the bacteriocins it produces [15]. Bacteriocin constitutes a special defense mechanism of bacteria, which can inhibit the growth of other bacteria and enable the bacteriocin-producing bacteria to gain a competitive advantage, which is of great significance for maintaining the balance of flora in the ecological environment.

Bacteriocin can inhibit the food-borne pathogen *Listeria monocytogenes*, and has been applied in food processing to improve food quality and safety, which has produced huge economic and social benefits [16]. At present, nisin is the most widely researched and applied bacteriocin. Nisin was first introduced to the market in the United Kingdom in 1953, and more than 50 countries have successively approved the application of nisin. In 1969, the Expert Committee of the Food and Agriculture Organization of the United Nations and the World Health Organization unanimously approved the use of nisin as a safe additive in food [17]. The successful development of nisin into a commercial product model greatly stimulated research on bacteriocin. In the future, enterocins may bring huge application value and economic benefits in food, feed addition or other aspects. This review will provide readers with important information on enterocins in terms of their biological properties, structure–function relationships, antibacterial mechanisms and applications in food systems, which will help in mining effective enterocins for various pathogenic bacteria and exploring the intervention method of “bacteria treatment with bacteria”.

## 2. Classification of Enterocins

In 1993, Klaenhammer proposed the first classification method of lactobacillus bacteriocin [18]. With the complexity of the bacteriocin structure and the continuous expansion of the bacteriocin family, the classification scheme has been continuously revised. At present, the latest and most authoritative three-category classification method has replaced the previous four-category classification scheme [19]. The classification standard of this classification method is based on the main structure, molecular weight, post-transcriptional modification and other biosynthetic mechanisms and biological activities of bacteriocin.

Class I bacteriocins refer to small-molecule (<10 kDa) thermostable bacteriocins with extensive post-translational modifications. Enzymatic modifications during biosynthesis, such as dehydration to lanthionine, glycosylation, head-to-tail cyclization and others [20], provide such bacteriocins with uncommon amino acid structures or structures that affect their activity. Nisin is a representative of class I bacteriocins. Cytolysin and enterocin AS-48 [21] are common class I enterocins. Cytolysin, which belongs to class Ia (lanthipeptides), are encoded by hemolytic virulence factors, so they do not have potential for food and drug applications [22]. Enterocin AS-48 was the first circular enterocin that was described and then classified into class Ib. Its N-terminal and C-terminal are bound by a peptide bond formed between the N-terminal methionine and the C-terminal tryptophan. Enterocin AS-48 has a total of 70 amino acid residues, containing no modified amino acid residues or disulfide bonds [23]. The secondary and tertiary structure of enterocin AS-48 has been reported. The structure of enterocin AS-48 consists of a spherical array of five α-helices surrounding a compact hydrophobic core [16]. The round shape and compact creases provide significant stability to the enterocin AS-48 molecule against extreme pH, heat and denaturants. Therefore, enterocin AS-48 has been widely applied to food systems due to its stability. Enterocin F4–9 belongs to class Ie (glycocins), containing glycosylated amino acid residues. Enterocin F4–9 does not completely kill the indicator bacteria but inhibits their growth [24], and the mechanism of action is unclear.

Class II bacteriocins refer to unmodified small-molecule (<10 kDa) thermostable bacteriocins after translation, which do not involve the formation of uncommon amino acids. Therefore, they do not require special enzymes other than signal peptides or transporters to complete the maturation and activation of such bacteriocins. Most enterocins are class II and these bacteriocins can inhibit the growth of *L. monocytogenes* [25]. Furthermore, most of them contain a double glycine signal peptide sequence between amino acid residues 15 and 30 of the N-terminal [26]. Enterocin A was a representative bacteriocin of enterocins and was the first enterocins of class IIa to be isolated, purified and relatively well-studied. The presence of the *entA* gene was detected in the first *E. faecium* strain *E. faecium* DO to complete whole-genome sequencing, but it did not exhibit bacteriocin activity. Enterocin X is a class IIb (two-peptide) bacteriocin derived from *E. faecium* KU-B5, and is composed of two antibacterial peptides (Xα and Xβ). Xα and Xβ are two peptides with different properties, which can produce antibacterial activity when they are mixed in equal proportions [27]. Class IIc enterocins (a type of N-terminal leaderless peptide) are composed of homologous peptides, encoded by tandemly repeated genes, and these peptides have a high homology of more than 70% in amino acid sequences. Enterocin L50 is a plasmid-encoded broad spectrum bacteriocin composed of enterocin L50A and enterocin L50B. These peptides share 72% sequence identity and are composed of 44 and 43 amino acids, respectively. Both of the two enterocins have antibacterial activity on their own, with entL50A being the most active. In addition, significant synergistic effects were observed when the two enterocins were used in combination, especially with specific indicator strains [28]. Enterocin B belongs to class IId and shows strong homology to carnobacteriocin A [29].

Class III bacteriocins include unmodified macromolecular heat-labile bacteriocins with molecular weights greater than 10 kDa. They have bacteriolytic activity and are the most poorly-characterized class of bacteriocins so far. Enterolysin A belongs to class III and has a bacteriolytic mode of action. Mature Enterolysin A is composed of 316 amino acids with a calculated molecular weight of 34,501 [30].

Although the taxonomic status of the genus *Enterococcus* was only established in 1982, many *Enterococcus* bacteriocins have been isolated and purified in recent years, and most of them were obtained from *Enterococcus faecalis* and *Enterococcus faecium*. According to the latest classification standard of bacteriocins, the main categories and properties of enterococcal bacteriocins are listed in Table 1.

## 3. Synthesis of Enterocins

Enterocins, produced by ribosome, have multiple functions in the innate immunity of bacteria and in microbial interactions [38]. In general, gram-negative bacteriocins are large-domain protein toxins with receptor-mediated narrow spectrum activity, while gram-positive bacteriocins are predominantly peptides [39]. Therefore, most enterocins are plain peptides, which may or may not contain disulfide bonds and cysteines. Genes that produce active enterocins are usually located in operon clusters, including plasmids, transposons and chromosomes [20]. The expression of these operons is induced by the auto-inducer peptide and regulated by a two-component control system [40] or a three-component control system [41].

Different classes of enterocins have different biosynthetic pathways (Figure 1) [42]. For class I lanthipeptides, the serine and cysteine residues in their propeptides undergo extensive post-translational modification. The lanthipeptides biosynthesis operons’ clusters contain genes including LanA (structural gene), LanB and LanC (encoding enzymes required for modification), LanT (encoding transport proteins), LanP (encoding a serine proteinase that is capable of removing the leader sequence), LanI and LanEFG (immunity gene). Their biosynthetic pathway follows the general scheme shown below: (1) formation of propeptides; (2) modification reactions; (3) proteolytic activation of the leader peptide; and (4) translocation of the mature or modified propeptides across the cytoplasmic membrane. Class II enterocins, however, do not undergo extensive post-translational modification. Propeptides are removed and exported out from the cell after the formation.

Once the enterocin is synthesized, it exists in an inactive form containing a leader peptide at the N-terminus. The leader sequence is an important factor to maintain the inactivity of the enterocins in the producing strain and can promote the interaction between the bacteriocin and the transporter. Gene clusters encoding leader peptides generally include structural, regulatory, transport and related modified genes. The leader peptide chain is usually cleaved through the transport system or the general secretion system of the cell. As more enterocins have been discovered and studied, several enterocins have been confirmed to be N-terminal leaderless peptides, such as enterocin L50, enterocin Q, enterocin EJ97 and enterocin RJ-11. These enterocins are divided into class IIc, as mentioned above.

## 4. Production and Purification of Enterocins

Enterocin genes usually have a low expression in wild strains. To improve the production of enterocins for better utilization, many researchers have investigated the production of recombinant proteins expressed by heterologous enterocins. Currently, three different heterologous expression methods have been proven to be effective: *E. coli*, yeast and *Lactobacillus* [44]. We sum up some common enterocins that have been heterologously expressed in three different systems (Table 2). Generally, heterologous expression in the above systems is efficient because these strains have relatively clear and mature genetic backgrounds that are convenient in terms of controlling the gene expression and achieving low culture costs with higher production of proteins of interest. Besides, chemical synthesis is also a great alternative way to attain pure enterocins without the complex purification of microbial fermentation.

The purification of enterocins is critical for the further study and application of enterocin after the enterocin has been produced. Purification methods can be divided into three categories according to the quality of the purified enterocins, which are crude extraction, moderate extraction and high-purity extraction. The crude extraction methods include ammonium sulfate precipitation, organic solvent extraction and adsorption. This method is suitable for the extraction of enterocin with unknown physical and chemical properties. The moderate purification methods can remove the impurity proteins in the crude enterocins, and include gel-filtration chromatography, ion-exchange chromatography and hydrophobic-interaction chromatography. Besides, high-purity extraction that involving reversed-phase HPLC and electrophoresis can produce enterocins with a purity over 95%, which meets the requirements of mass spectrometry and amino acid sequencing.

## 5. Antibacterial Activity and Mechanisms of Action

Enterocins are polypeptides or precursor polypeptides with bacteriostatic activity produced by *enterococcus* during metabolism. Previous studies have shown that enterocins exhibit strong antimicrobial activity against important foodborne pathogens, including *Staphylococcus aureus*, *Listeria monocytogenes* and *Salmonella enterica* et al. [53]. For example, purified enterocin E-760 has been proven to inhibit the growth of 24 species of gram-negative bacteria and 3 species of gram-positive bacteria. The MIC of enterocin E-760 for these indicator bacteria ranges from 0.1 μg/mL to 3.2 μg/mL [57].

The antibacterial mechanism of bacteriocins, which differs from the antibiotics, is closely connected with the classification of bacteriocins and the indicator bacteria [58]. It is crucial for the normal cell to maintain the integrity of cell walls and membranes. Bacteriocin inhibits other bacteria by perforating on target cells, inhibiting peptidoglycan synthesis, inhibiting protein synthesis by interacting with ribosomes or tRNA, and directly degrading target cell DNA [59,60].

The antibacterial mechanisms of enterocins can be classified into the cell envelope and inhibition of gene expression (Figure 2). Several enterocins can interact with specific bacterial cell receptors, including lipid II and other related cell wall precursors, undecaprenyl pyrophosphate phosphatase, the mannose phosphotransferase system, the maltose ABC transporter or a Zn-dependent metallopeptidase [61]. In addition, most class II enterocins can induce membrane permeabilization and then cause the leakage of target cells. Enterocin P is an amphipathic and cationic peptide, which was proven to interact with the negatively charged bacterial membranes of the target cells and then cause death [62]. The majority of class III enterocins, which are large and heat-labile proteins, can break down the sensitive cells by catalyzing cell-wall hydrolysis. Enterolysin A can hydrolyze peptide bonds in the peptidoglycan of target cells [30]. Besides, enterocins can control their target bacteria by interfering in DNA replication and mRNA synthesis and transcription [63].

## 6. Application of Enterocins in Food Industry

Many enterocins have been found to be active against pathogens and used as food preservatives for food protection. Examples of applications of enterocins in three main food systems are listed in Table 3.

### 6.1. Dairy Products

Milk is a highly nutritious food and provides a perfect environment for a large variety of microorganisms [64]. Dairy products are easily contaminated by various pathogenic microorganisms in the process of production, packaging, transportation and sales, which is considered to be a great threat to the quality and safety of food. *Staphylococcus aureus* and *Listeria monocytogenes* are common foodborne pathogens and have caused food poisoning outbreaks many times [65]. Enterocins in dairy products can naturally prevent the growth of microorganisms.

Enterocin AS-48, produced by *E. faecalis A-48-32*, is a circular bacteriocin with a wide-range bactericidal activity against various gram-positive bacteria, such as *S. aureus*, *Bacillus cereus* and *L. monocytogenes* [23,66]. The effectiveness of enterocin AS-48, which controls *S. aureus* and *B. cereus* in skim milk and fresh cheese, has been determined. Enterocin AS-48 combined with moderate heat treatment (65 °C, 5 min) showed a strong anti-staphylococcal synergy at a concentration of 20 μg/mL, and was able to remove staphylococci after six hours of incubation [67]. In another research study, Arantxa tested enterocin AS-48 against *B. cereus* in milk and cheese. The results of the experiments revealed that no viable *B. cereus* cells were found after 72 h incubation in food systems that were inoculated with *B. cereus* and produced the strain of enterocin AS-48.

In addition, Lauková with his lab members were the first to detect enterocin CCM 4231 produced by *E. faecium CCM 4231*, isolated from the rumen content of a calf [68]. They carried out tests for the antibacterial activity of this enterocin against different pathogens especially in dairy products. In skim milk, live cells of the *S. aureus* decreased from 10^10^ cfu/mL to 10^2^ cfu/mL 24 h after the addition of enterocin CCM 4231, but the control did not show a decrease in staphylococcal count (enterocin was not added). During the preparation of yogurt, significant inhibition of the *S. aureus* was observed after the addition of enterocin CCM 4231 for 3.5 h [69]. Besides, inhibition of enterocin CCM 4231 (3200 AU/mL) was found in experimental cheese samples via the addition of *Listeria* strains during the preparation of St. Paulin’s cheese, from an initial concentration of 6.7 ± 0.06 log cfu/g to 1.9 ± 0.01 log cfu/g after one week, compared to a control treatment with pathogen levels of 6.0 ± 0.01 log cfu/g after one week [70]. Similarly, in other studies, many enterocins were also proven to be inhibitory to pathogens, including enterocin EJ97 [35], enterocin E-760 [57] and enterocin CRL 35 [71]. These above results clearly demonstrate the antibacterial potential of enterocins, which have great application prospects in the processing of dairy products.

### 6.2. Meat Products

In addition to dairy products, other products of animal origin, namely meat and fish, are also very beneficial to the growth of microorganisms due to their moisture and rich nutrients. Foods of animal origin are the major hosts for many foodborne pathogens, including *Salmonella*, *Campylobacter*, *L. monocytogenes* and the *Shiga* toxin-producing strains of *Escherichia coli* [72]. According to the data collected by the European Food Safety Authority (EFSA), a total of 5196 weak and strong outbreaks involving meat products were reported in Europe alone in 2013. In these outbreaks, *Salmonella enterica* remained the most common pathogen (22.5%), followed by viruses (18.1%) and bacterial toxins (16.1%) [73]. The potential for zoonotic diseases and their ability to produce enough toxins to cause illness and even death illustrates the seriousness of the situation. Food poisoning causes millions of sporadic illnesses, chronic complications, and even large-scale and difficult outbreaks across many countries and nations. Therefore, the use of enterocins has potential in the preservation of meat.

The anti-listeria effect of enterocins A and B, produced by *E. faecium CTC492*, has been demonstrated in meat and meat products (cooked ham, chicken breast and fermented sausages). Enterocins A and B (4800 AU/g) diminished the number of *Listeria* by 7.98 log cycles in cooked ham. In chicken breasts stored at 7 °C for 7 days, 4800 AU/cm^2^ enterocins A and B reduced the number of *Listeria* by 5.26 log cycles compared to the control [74]. After that, a number of researchers combined the application of enterocins A and B and high hydrostatic pressure (HHP) to extend the shelf-life of food products. Enterocins were applied to cooked and dry cured ham with *L. monocytogenes* and *Salmonella* and subjected to high-pressure treatment at 600 MPa. *Salmonella* and *L. monocytogenes* were not detected at 25 g in both cooked and dried prosciutto and remained at this level during the entire storage [75].

The potential of enterocin AS-48 to inhibit food-derived pathogens in meat products has been demonstrated. No viable *L. monocytogenes* was detected after culturing at 20 °C for 6 and 9 days in a meat sausage model system supplemented with enterocin AS-48 [76]. In the cooked ham model system, enterocin AS-48 alone (20, 40, and 60 μg/g) was active against *L. monocytogenes* at 5 and 15 °C, but it could not avoid the regrowth of *Listeria* during the 60 days of shelf-life. However, enterocin AS-48 combined with chemical preservatives or heat can improve the efficacy of anti-*Listeria* during storage at 5 °C. The most effective combination was AS-48-nitrite/nitrate (0.007%), which caused *Listeria* to fall below detection levels from the start to the end of storage [77]. As with the enterocins A and B, enterocin AS-48 combined with HHP can inhibit the growth of *S. enterica*, *L. monocytogenes* and *S. aureus*. In another research study, 148 AU/g enterocin AS-48 caused a dramatic reduction of 5.5 log cfu/g in *L. monocytogenes* and a significant inhibition for *Salmonella* at the end of ripening [78]. Enterocin LM-2 was also combined with HHP to suppress the growth of *Salmonella* and *L. monocytogenes* [79]. Besides, enterocin CCM 4231 was reported to be applied to inhibit *L. monocytogenes* in dry fermented Hornád salami, resulting in a 1.67 log cycle reduction in *L. monocytogenes* [80].

### 6.3. Fruits and Vegetables Products

Fresh fruits and vegetables are an important part of the human diet, and public health agencies recommend eating at least five servings of fruits and vegetables daily. These fruits and vegetables are rich in minerals and vitamins, along with trace elements such copper, manganese and zinc, which are components of various enzyme cofactors required for different metabolic reactions. There has been a growing demand for fresh and healthy food, especially fruits and vegetables, during the last decade. However, the consumption of fruit and vegetable products is also susceptible to contamination by pathogens. Since vegetables and fruits are organisms with different compositional properties, their microbial fauna can vary significantly depending on the pH, water activity and availability of nutrients [81]. Mesophilic enteric pathogens such as *E. coli*, *Salmonella* and *L. monocytogenes* are common pathogens that cause disease in humans. *Salmonella* and *E. coli O157: H7* can cause infection even at low doses and grow very rapidly [82]. Enterocins have been investigated in fruits and vegetables in relation to the preservation of such products.

Plant-derived foods have been studied in detail as targets for the application of enterocin AS-48. When enterocin AS-48 (6 μg/mL) is added to low-acid canned vegetables (tomato paste, canned peach syrup and canned pineapple juice), *Bacillus coagulans* cells are completely or partially inactivated [83]. Enterocin AS-48 is effective in inactivating *L. monocytogenes*, *B. cereus* and *Bacillus weihenstephanensis* in sprouts, as well as *L. monocytogenes* in sliced fruits and whole fruit pieces when used as washing treatment alone or in combination with other chemical antimicrobials [84,85,86]. Enterocin AS-48 can be a good candidate to apply in the preservation of fruits juices because of the strong stability of this enterocin in juices [87]. When it was added to juices contaminated with endospores of *Alicyclobacillus acidoterrestris* with low concentrations, the bacteria could be inhibited completely for up to two weeks in fresh apple and orange juices, and for up to two to three months in various commercial fruit juices under storage temperatures [88]. Widespread enterocin AS-48 was tested for biopreservation against aerobic mesophilic endospore-forming bacteria in ready-to-eat vegetable foods (soups and purees). *B. cereus LWL1*, in six vegetable products tested (natural vegetable cream, asparagus cream, traditional soup and homemade traditional soup), was inhibited totally by adding AS-48 (10 μg/mL) for up to 30 days at 6, 15 and 22 °C [89].

As with the production of other food products, HHP was combined with enterocin AS-48 in the fruits and vegetables. As is shown in previous research studies, the combination of AS-48 and HHP was more effective in maintaining the quality of cherimoya pulp than HHP treatment alone during storage [90].

Enterocin 416K1, produced by *E. casseliflavus IM 416K1*, demonstrated strong anti-*listerial* activity with artificially contaminated *L. monocytogenes* ready-to-eat vegetables and fruits [91]. The research showed that in apples and grapes, the number of viable *Listeria* in the control dropped to about 2 log cfu/g within one day and remained constant until the experiment ended. In the treated sample, *L. monocytogenes* was completely inhibited. This was achieved within the first hour of using chitosan and enterocin together, within 8 h for the enterocin-only sample, and within 72 h for the chitosan-only sample. Besides, enterocin EJ97 can control the vegetative cells and spores of *Geobacillus stearothermophilus*, which causes sour spoilage of high-water-activity and low-acid canned foods [92].

Most enterocins show a certain degree of bacteriostatic activity. However, a number of enterocins only show strong bacteriostatic activity when they act synergistically with other substances. Enterocin KT2W2G, combined with cinnamon oil at 4:6 ratios, exhibited the strongest synergistic effect spoilage microorganisms (*Lactococcus lactis*, *E. faecalis*, *Klebsiella pneumonia*, *Serratia marcescens pichia* and *Candida metapsilosis*), whereas enterocin KT2W2G displayed no activity when using alone [93].
molecules-27-02258-t003_Table 3Table 3Application of enterocins in food systems.Food SystemsEnterocins Producing StrainAdditional TreatmentTarget OrganismsReferencesDairy productsEnterocin AS-48*E. faecalis A-48-32*moderate heat (65 °C, 5 min); Cocultivation;*Staphylococcus aureus;**Bacillus cereus*[23,92]Enterocin CCM 4231*E. faecium CCM 4231*add purified enterocin CCM 4231 (concentration 3200 AU/mL) *Escherichia coli;**Listeria monocytogenes;**Staphylococcus aureus*[68,78]Enterocin EJ97*E. faecalis EJ97*sodium nitrite, sodium benzoate, et.al*Listeria monocytogenes*;[35]
Enterocin E-760*E. faecium NRRL B-30745*
*Salmonella enterica;**Yersinia enterocolitica;*[54]Dairy productsEnterocin CRL 35*E. faecium CRL 35*
*Listeria monocytogenes*;[69]Meat productsEnterocins A and B*E. faecium CTC492*
*Salmonella entericai; Listeria monocytogenes; Staphylococcus aureus;*[72,93]Enterocin AS-48*E. faecalis A-48-32*added at concentrations of 30 or 40 μg/g;high hydrostatic pressure*Staphylococcus aureus; Salmonella enterica;**Listeria monocytogenes;*[76,94]Enterocin CCM 4231*E. faecium CCM 4231*
*Listeria monocytogenes*;[78]Enterocin LM-2*E. faecium LM-2*high hydrostatic pressure*Salmonella enterica;**Listeria monocytogenes;*[77]Fruits and vegetables productsEnterocin AS-48*E. faecalis A-48-32*Washing with chemical preservatives; heat (80–95 °C for 5 min); nisin and phenolic compounds; cinnamic and hydrocinnamic acids, etc*Bacillus cereus;**Listeria monocytogenes;**Staphylococcus aureus; Bacillus coagulans; Paenibacillus sp.; Bacillus weihenstephanensis; Pantoea; Leuconostocs*[81,83,86,88,95]
Enterocin 416K1*E. casseliflavus IM 416K1*chitosan*Listeria monocytogenes*[89]Fruits and vegetables productsEnterocin EJ97*E. faecalis EJ97*sodium tripolyphosphate and sodium nitrite*Bacillus coagulans*[96]Enterocin KT2W2G*E. faecalis KT2W2G*essential oils; cinnamon oil*Lactococcus lactis subsp. Lactis; Enterococcus faecalis; Klebsiella pneumonia; Klebsiella variicola; Serratia marcescens*[91]


## 7. Conclusions

In recent years, there has been a growing interest in the use of bacteriocins to partially replace traditional antibiotics in humans or animals. Bacteriocin has the characteristics of high efficiency, high specificity, low toxicity, less drug-resistance, etc. It meets food safety grade requirements and is widely used in the food industry to improve food quality and safety, which has produced huge economic and social benefits. So far, nisin is the only bacteriocin approved for use as a food preservative in more than 50 countries. Various tests have proved that nisin has extremely high safety. Its median lethal dose (LD50) to mice is about 7000 mg/kg, which is close to the LD50 of ordinary table salt. Enterocins, as a bacteriocin, have very broad prospects in food and medical applications.

Although many patents on enterocins have been reported [94,95,96,97], these enterocins still have limitations. The moderate antimicrobial effects and limited specificity are major limitations in the application of enterocins in the food industry [98]. Additionally, certain enterocins may be inactivated by food components, which is an important problem and it is almost impossible to find the exact cause of the inhibition [87]. It has been nearly a century since the discovery and utilization of bacteriocins; however, the research on enterocins is still mainly limited to the aspects of mining and application. Although several mechanisms of action have been reported, they mainly focus on the target and regulation of action. There is still a lack of deeper and more comprehensive research, and an in-depth understanding of the classification, expression and mechanisms would be beneficial in terms of the large scale production and wide application of enterocins.

## Figures and Tables

**Figure 1 molecules-27-02258-f001:**
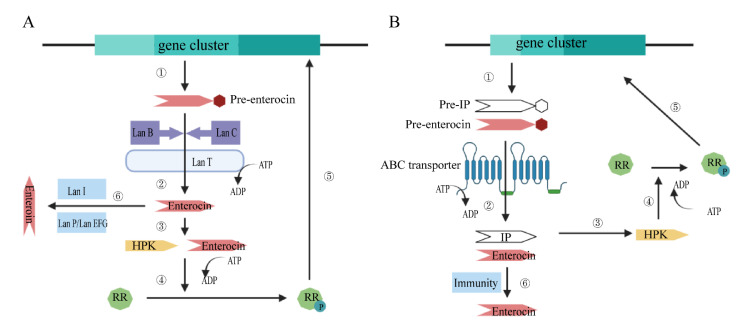
(Adapted from [43]) Schematic Diagram (**A**)**:** The biosynthesis pathway of several class I enterocins (lanthipeptides): (1) formation of preenterocin; (2) modification of preenterocin by LanB and LanC and translocation of preenterocin by LanT; (3) histidine protein kinase (HPK) senses the presence of enterocin and autophosphorylates; (4) the phosphoryl group (P) is transferred to the response regulator (RR); (5) the transcription of regulated genes is activated by RR; (6) mature enterocin is processed and released under the regulation of LanP, LanI and Lan EFG. Schematic Diagram (**B**)**:** The biosynthesis pathway of class II enterocins: (1) formation of the preenterocin and prepeptide of the induction factor (IF); (2) the precursors are processed and translocated by the ABC transporter and then mature enterocin and IF are released; (3) HPK senses the presence of autophosphorylates and IF; (4) the phosphoryl group (P) is transferred to the response regulator (RR); (5) the transcription of regulated genes is activated by RR; (6) producer immunity is achieved.

**Figure 2 molecules-27-02258-f002:**
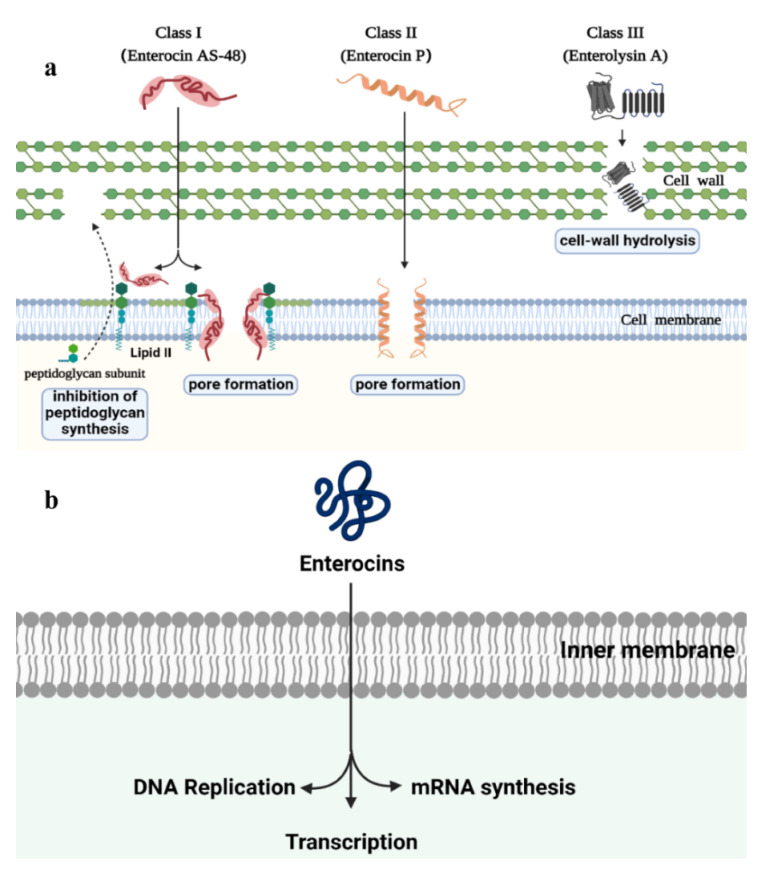
The antibacterial mechanism of enterocins (adapted from [59]). Enterocins can be divided into different groups according to their mode of action: (**a**) Several class I enterocins have two antimicrobial methods. They can bind to lipid II and prevent peptidoglycan synthesis. Furthermore, they can also use lipid II to initiate membrane insertion and form pores. Class II enterocins generally have an amphiphilic helical structure, allowing them to be inserted into the cell membrane and causing the death of target cell. Class III enterocins, such as Enterolysin A, can degrade the sensitive cells by hydrolyzing the specific peptide bonds of cell wall. (**b**) A number of enterocins control the target bacteria by interfering with gene expression.

**Table 1 molecules-27-02258-t001:** Classification of Enterocins.

Classification		Characters	Examples
Class I	I a	posttranslationally modified; lanthipeptides	Cytolysin [31]
	I b	posttranslationally modified; head-to-tail cyclized peptides;	Enterocin AS-48 [16]
	I c	posttranslationally modified; sactibiotics	
	I d	posttranslationally modified; linear azol(in)e-containing peptides	
	I e	posttranslationally modified; glycocins	Enterocin F4-9 [32]
	I f	posttranslationally modified; lasso peptides	
Class II	II a	unmodified bacteriocins; pediocin-like bacteriocins	Enterocin A [25]; Enterocin P [33]
	II b	unmodified bacteriocins; two-peptide bacteriocins	Enterocin X [27]
	II c	unmodified bacteriocins; leaderless bacteriocins	Enterocin L50 [28]; Enterocin Q [34]; Enterocin EJ97 [35]; Enterocin RJ-11 [36]
	II d	unmodified bacteriocins; non-pediocin-like, single-peptide bacteriocins	Enterocin B [29]
Class III		large molecular weight; heat labile	Enterolysin A [37]

**Table 2 molecules-27-02258-t002:** Heterologous expression of enterocins in different systems.

Expression Systems	Enterocins	Producing Strain	Expression Host	Plasmid	References
*E. coli*	Enterocin P	*E. faecium* P13	*E. coli* ER2566	pTWIN1	[45]
	Enterolysin A	*E. faecalis* II/1	*E. coli* SG13009	pQE-30 UA	[46]
	Enterocin CRL35	*E. mundtii* CRL35	BL21(DE3)	pACYCDuet-1	[47]
	Enterocin A and B	*E. faecium* ATB197a	BL21(DE3)	pET37b(+)	[48]
Yeast	Enterocin HF and enterocin CRL35	*E. faecium*	*P. pastoris* X-33	pPICZαA	[49]
	Enterocin A	*E. faecium* CTC492; *E. faecium* T136	*P. pastoris* X-33; *K. lactis* GG799EA	pPICZαA; pKLAC2	[50,51]
	L50A and L50B	*E. faecium* L50	*P. pastoris* X-33	pPICZαA	[52]
	Enterocin P	*E. faecium* P13	*P. pastoris* X-33	pPICZαA	[53]
*Lactobacillus*	Enterocin A	*E. faecium* T136; *E. faecium* PLBC21	*L. casei* CECT475; *L. casei* IL1403; *L. casei* NZ9000	pSIP411UAI; pMG36c	[54,55]
	Ent53B	*E. faecium* NKR-5-3	*L. casei* NZ9000	pNK-B	[56]

## Data Availability

Data sharing not applicable. No new data were created or analyzed in this study.

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
