# Peer review of "Enterocins: Classification, Synthesis, Antibacterial Mechanisms and Food Applications"

_molecules, 2022, doi:10.3390/molecules27072258_

Round 1

Reviewer 1 Report

The paper discusses an interesting, but also already well described subject in a comprehensive way and is well written. The language is good, but there are aspects of the discussion that need some detail or supportive data, or some other aspects that have not been discussed adequately so far (like the production and purification process of enterocins). Below are my comments to the authors in more detail.

  1. Table 1. Reference is needed.
  2. Synthesis of Enterocins (p. 5). A more detailed description of biosynthetic pathways including a diagram of the biosynthetic pathway and the enzymes involved would be important.
  3. Before chapter 5 (applications of enterocins in food industry), it would be useful to have a previous chapter with in vitro tests of antimicrobial activities of enterocins. These in vitro results of antibacterial activity could be also inserted into chapter 4, (which could rephrased as Antimicrobial activity and Mechanism of Action).
  4. Page 7 and Table 2. “Arantxa” (if it spelled correctly seems to be the fist name of an author, which is not necessary. What we need for the reference is the surname, which is Munoz.
  5. Table 2 needs a better alignment of the columns and line, especially the column of “additional treatment”, as it is not clear to which line it refers to.
  6. There is at least one more chapter that would be necessary for such as review, and this the (industrial or lab scale) Production and Purification of Enterocins. In such a chapter the authors can explain how these molecules can be produced and purified before being used in food and other applications.
  7. I would like to see some data regarding patents on Enterocins (and other bacteriocins) and some discussion and explanation about the reason why only nisin has been produced and applied industrially so far. What are the limitations for the industrial production and application of enterocins? Why haven’t they made their way to the market so far?

Reviewer 2 Report

The present review discusses enterocins, their classification, synthesis, antibacterial mechanisms and applications, and gives some views for future research.

The work is well organized, and it is interesting and timely prepared.

Minor points to be addressed:

  • This reviewer thinks that it should be interesting to specifically say “ESKAPE microorganisms” in lines 26-28;
  • Some references need to be formatted (“[2][7]”);
  • It is important to add the reference for each case in Table 1;
  • References of table 2 are not well formatted;
  • “leuconostocs” is not well written (capital letter lacking).
